

# Effect of temperature on pollen germination for several Rosaceae species: influence of freezing conservation time on germination patterns

Roberto Beltrán[1], Aina Valls[1], Nuria Cebrián[1], Carlos Zornoza[1], Francisco García Breijo[1], José Reig Armiñana[2], Alfonso Garmendia[3] and Hugo Merle[1]

[1] Departamento de Ecosistemas Agroforestales, Universitat Politècnica de València, Valencia, Spain
[2] Instituto Cavanilles de Biodiversidad y Biología Evolutiva, Universidad de Valencia, Valencia, Spain
[3] Instituto Agroforestal Mediterráneo, Universitat Politècnica de València, Valencia, Valencia, Spain

## ABSTRACT

Between February 2018 and April 2018, flowers were collected from eight Rosaceae species. Flowers were kept in a freezer at −20 °C for three freezing times (Treatment 1, two months; Treatment 2, four months; Treatment 3, six months). After extracting pollen, *in vitro* germination was induced in a culture medium and incubated at six different temperatures for 72 h. The percentage of pollen germination, average pollen tube length and maximum pollen tube length were measured. Pollen germination was maximum for all species between 15 °C and 30 °C. *Cydonia oblonga*, *Malus sylvestris*, *Prunus avium*, *Prunus domestica*, *Prunus dulcis*, *Prunus persica* and *Pyrus communis* obtained 30–52% pollen germination between 15 °C and 20 °C. *Prunus cerasifera* had 40% pollen germination at 30 °C. All species studied reached the maximum pollen tube length between 10 °C and 25 °C. Germination did not change significantly for any of the species with freezing time, but we found significant differences in the three parameters measured between treatments. The highest germination percentages were obtained in Treatment 2 (four months frozen at −20 °C), while the maximum pollen tube length was reached in Treatment 1 (two months frozen at −20 °C). According to our results, freezing time affected the germination-temperature patterns. This could indicate that studies on the effect of temperature on pollen germination should always be carried out with fresh pollen to obtain more conclusive data.

Corresponding author
Roberto Beltrán,
robelmar@upvnet.upv.es

## INTRODUCTION

All Rosaceae species present some common characteristics, like actinomorphic and hermaphroditic flowers requiring pollination for fruit formation. This is the case of the subfamilies *Maloideae* (apples, pears, quinces) and *Prunoideae* (almonds, cherries, peaches, plums), which depend on fertilization for fruit set (*Hegedűs, Lénárt & Halász, 2012*). Specifically, *Malus sylvestris* (L.) Mill. and *Pyrus communis* L. are allogamous species that require the presence of pollinators (e.g., bees or bumblebees) for obtaining higher

yields (*Stern et al., 2004*). Other species such as *Prunus avium* (L.) L., *Prunus domestica* L. and *Prunus persica* (L.) Batsch present autogamous (self-pollinated) and allogamous varieties. In the case of the self-pollinated ones, the yield is related to the fruit set, sometimes requiring the application of hormones to improve it (*Webster & Goldwin, 1981*; *Goldwin & Webster, 1983*). In both cases, the pollen grain must be deposited on the stigma and emit the pollen tube. Fertilization occurs when the pollen tube reaches the ovule (*Monselise, 2017*). Mature ovules have a limited lifetime, so fertilization must occur throughout this time, also called the effective pollination period (*Sanzol & Herrero, 2001*). It is a complex process, where temperature plays a significant role. It has been observed that an increase in temperature produces both a faster growth of the pollen tube, and an acceleration of the ovule degeneration (*Hedhly, Hormaza & Herrero, 2005*). Changes in temperature can alter the effective period of pollination, which in turn determines the final fruit set and the yield that the variety can obtain (*Sanzol & Herrero, 2001*). The weather determines many of the pollination parameters of these species, such as the pollen germination percentage. Thus, it is important to know what the best pollen germination conditions are because reduced germination of pollen may represent important-limiting variables for the crop (*Weinbaum, Parfitt & Polito, 1984*). Likewise, the presence of seeds is decisive for the development of the fruit, so fertilization is one of the processes that must be followed with greater attention in these crops (*Monselise, 2017*).

The main Rosaceae species come from very diverse origins. *Malus*, *Prunus* (*Pr*) and *Pyrus* (*Py*) genera come from areas with a temperate climate (*Asghar, Ali & Yasmin, 2012*; *Castede et al., 2014*; *Silva et al., 2014*). On the contrary, other Rosaceae species are typical of areas with a warmer climate, like *Cydonia oblonga* Mill. and *Prunus cerasifera* Ehrh. (*Hegedűs & Halasz, 2006*; *Postman, 2009*). Due to this, the temperature ranges that each species requires throughout the flowering and fruiting phases are variable (*Monselise, 2017*). Climate change can become a limiting factor for some crops in many areas of the planet (*Saxe et al., 2001*). The global average temperature increase can affect certain regions to a greater extent. In this sense, *Iglesias et al. (2012)* indicated that a change in European crop patterns is expected, with a decrease in total productivity in Southern Europe and an increase in productivity in Northern Europe. The rise in temperatures could benefit some crops but harm many others. In some areas of the Mediterranean Basin, the progress of crops with higher thermal requirements is already observed. As an example, in some areas of southern Spain such as the provinces of Murcia and Alicante, we can observe at the moment avocado fields. Avocado is a typical crop of warmer areas that is now possible to cultivate in northern latitudes like ours (personal observation). On the contrary, other crops that require cold temperatures could be altered. In these cases, excess heat during flowering and fertilization determines the harvest yield (*Hedhly, Hormaza & Herrero, 2005*). This could involve the withdrawal to more northern areas of these crops. For all these reasons, knowing the temperature limits of these processes can become an interesting tool to detect possible problems and know if there are species more sensitive than others to these threats.

To date, several studies have been published about the effect of temperature on pollen germination in several Rosaceae species, such as *Py. communis* (*Vasilakakis & Porlingis,*

*1985*), apricot (*Prunus armeniaca* L.) (*Egea et al., 1992*), *Pr. avium* (*Hedhly, Hormaza & Herrero, 2004*), Chinese plum (*Prunus mume* (Siebold) Siebold & Zucc.) (*Wolukau et al., 2004*) *Pr. persica* (*Hedhly, Hormaza & Herrero, 2005*), almond (*Prunus dulcis* (Mill.) D.A.Webb) (*Sorkheh et al., 2011b*) and various native Iranian almonds like *Prunus eleaegnifolia* Mill., *Prunus orientalis* Mill., *Prunus lycioides* Spach, *Prunus reuteri* Bioss et Bushe, *Prunus arabica* Olivier, *Prunus glauca* Browick and *Prunus scoparia* Spach (*Sorkheh et al., 2011a*). According to these studies, most of these species reach the maximum pollen germination percentage or the maximum length of the pollen tube at a temperature close to 20 °C. Recently, *Sorkheh et al. (2018)* carried out the pre-incubation of different *Pr. dulcis* varieties pollen at different temperatures, thus reproducing the conditions that pollen can face during its dispersion process. Some studies have also compared results among several species, such as *Weinbaum, Parfitt & Polito (1984)*, who indicated the different sensitivities of almond and peach pollen to low temperatures. Other studies have been carried out in different botanical family crops like groundnut (*Arachis hypogaea* L.; *Kakani et al., 2002*), different pepper species (*Capsicum* spp.; *Reddy & Kakani, 2007*), *Pistacia* spp. (*Acar & Kakani, 2010*), longan (*Dimocarpus longan* Lour.; *Pham, Herrero & Hormaza, 2015*) and coconut (*Cocos nucifera* L.; *Hebbar et al., 2018*).

Most studies about pollen germination have focused on reviewing the effect of temperature on germination. Reports that have dealt with the most appropriate methodology to produce *in vitro* pollen germination are less common. *Feijo, Malho & Obermeyer (1995)* and *Fan et al. (2001)* studied the importance of different ions, such as calcium or potassium, for pollen germination in several species. *Rosell, Herrero & Saúco (1999)* improved the pollen germination method in cherimoya (*Annona cherimola* Mill.) by specifying several aspects of the existing methodology (germination temperature, pre-hydration, etc.). *Wolukau et al. (2004)* investigated the effect of polyamines and a polyamine synthesis inhibitor on both pollen germination and pollen tube growth in *Pr. mume. Burke, Velten & Oliver (2004)* worked with cotton (*Gossypium* spp.) pollen and made several improvements in the existing pollen germination method.

As previously mentioned, a better understanding of the phenology of the crop with respect to temperatures can be of vital importance to precede possible scenarios of Climate Change in the future. The flowering season of the Rosaceae species cited in this study usually lasts a few weeks. In many cases, pollen germination studies cannot be carried out on fresh pollen, due to management and sample availability. It is therefore the case that many of these studies must be carried out with stored freezing pollen. Thus, to be able to carry out new studies and to be able to compare results, it will be very important to know how freezing affects pollen germination. Several authors have reported information on the viability of pollen stored at various temperatures. Many of these studies coincide in stating that the pollen storage at low temperatures like −20 °C does not result in a loss of its germination capability. So, *Perveen & Khan (2008)* studied pollen germination in *Malus pumila* Mill and indicated that pollen stored at low temperatures had better germination rates than fresh pollen or stored pollen at 4 °C. The same authors analyzed the effect of storage temperatures on pollen germination for species of other botanical families, such as *Abelmoschus esculentus* L. (*Khan & Perveen, 2006a*), *Solanum melongena* L. (*Khan &*

*Perveen, 2006b*), *Citrullus lanatus* L. (*Khan & Perveen, 2010*), *Lagenaria siceraria* (Molina) Standley (*Khan & Perveen, 2011*) and *Praecitrullus fistulosus* (Stocks) Pangalo (*Perveen & Khan, 2011*). In all cases, pollen samples stored at −20 °C and −30 °C showed a better pollen germination percentage than fresh pollen samples. Likewise, *Weinbaum, Parfitt & Polito (1984)* carried out their study of the effect of cold temperatures on *Pr. dulcis* and *Pr. persica* pollen germination with pollen stored at −20 °C. Nor was a loss in the germination capability of pollen observed in this study. Therefore, the storage of pollen at low temperatures (−20 °C, −30 °C or −60 °C) is a suitable method for long-term use of pollen.

In any case, we found no study about temperature storage conditions for pollen from the most important Rosaceae species, such as apple, cherry, pear or plum trees. For all this, our main research objectives were to check (i) pollen germination capacity at different temperatures, (ii) average and maximum pollen tube length at different temperatures and (iii) the influence of the freezing conservation time for these species.

## MATERIALS & METHODS

Eight Rosaceae species from different origins were selected. Apple (*M. sylvestris*) and pear (*Py. communis*) trees are species from temperate climates that better tolerate low temperatures. Both species need higher chilling requirement than other species in the same family (*Heide & Prestrud, 2005*). Quince (*C. oblonga*) and cherry plum (*Pr. cerasifera*) trees come from warmer climates and, therefore, show more tolerance to high temperatures. The other four tree species, cherry (*Pr. avium*), plum (*Pr. domestica*), almond (*Pr. dulcis*) and peach (*Pr. persica*), are of the intermediate climate origins between both groups. Between February 2018 and April 2018, flowers from all these species were taken at anthesis. Samples were collected at different locations in three Spanish provinces located at different altitudes above sea level (Table 1).

All the samples were kept in bags and stored in a freezer at −20 °C. The freezing times for all the species, were two, four and six months under the conditions mentioned above. According to this, three experimental treatments were established depending on the time when flowers were frozen: Treatment 1 (two months frozen), Treatment 2 (four months frozen) and Treatment 3 (six months frozen).

Flowers were taken from freezers when they were needed to perform assays. They were placed inside a humid chamber at 4 °C for 2 h before extracting pollen to achieve its pre-hydration (*Mesejo et al., 2006*). Pollen grains were extracted and placed in 5 ml of BK medium (a modification of BK medium that contained 100 g l$^{-1}$ sucrose, 0.1 g l$^{-1}$ H$_3$BO$_3$, 0.3 g l$^{-1}$ Ca (NO$_3$)$_2$, 0.1 g l$^{-1}$ KNO$_3$ and 10 g l$^{-1}$ agarose; *Brewbaker & Kwack, 1963*) to induce their germination. The content of two to four anthers per plate was introduced. Plates were incubated for 72 h in the dark at the same temperatures as those used in other studies: 5 °C, 10 °C, 15 °C, 20 °C, 25 °C and 30 °C (*Weinbaum, Parfitt & Polito, 1984*; *Hedhly, Hormaza & Herrero, 2004*).

To assess germination capacity, the percentage of germinated pollen grains average pollen tube length and maximum pollen tube lengths were used. It was considered that

Beltrán et al. (2019), *PeerJ*, DOI 10.7717/peerj.8195

**Table 1  Origin of the samples.** Coordenates and climate data for sampling locations.

| Botanical name | Common name | Variety | Location | Province | Latitude (N) | Longitude (E) | Altitude | Average temperature | Annual rainfall | Climate[a] |
|---|---|---|---|---|---|---|---|---|---|---|
| *Cydonia oblonga* | Quince | – | Montserrat | Valencia | 39.36044 | −0.54543 | 168 | 16.8 °C | 432 | BSk |
| *Malus sylvestris* | Apple | Golden | Casas-Ibáñez | Albacete | 39.28688 | −1.47133 | 707 | 13.8 °C | 417 | Csa |
| *Prunus avium* | Cherry | Ambrunes | Casas-Ibáñez | Albacete | 39.28688 | −1.47133 | 707 | 13.8 °C | 417 | Csa |
| *Prunus cerasifera* | Cherry plum | Pissardii | Valencia | Valencia | 39.46975 | −0.37739 | 16 | 17.4 °C | 445 | BSk |
| *Prunus domestica* | Plum | Santa Rosa | Montserrat | Valencia | 39.36044 | −0.54543 | 168 | 16.8 °C | 432 | BSk |
| *Prunus dulcis* | Almond | Marcona | Chóvar | Castellón | 39.85096 | −0.32002 | 415 | 14.8 °C | 483 | Csa |
| *Prunus persica* | Peach | Maruja | Casas-Ibáñez | Albacete | 39.28688 | −1.47133 | 707 | 13.8 °C | 417 | Csa |
| *Pyrus communis* | Pear | Blanquilla | Casas-Ibáñez | Albacete | 39.28688 | −1.47133 | 707 | 13.8 °C | 417 | Csa |

**Notes.**
[a] Köppen climate classification.
BSk, steppe climate; Csa, Mediterranean climate.

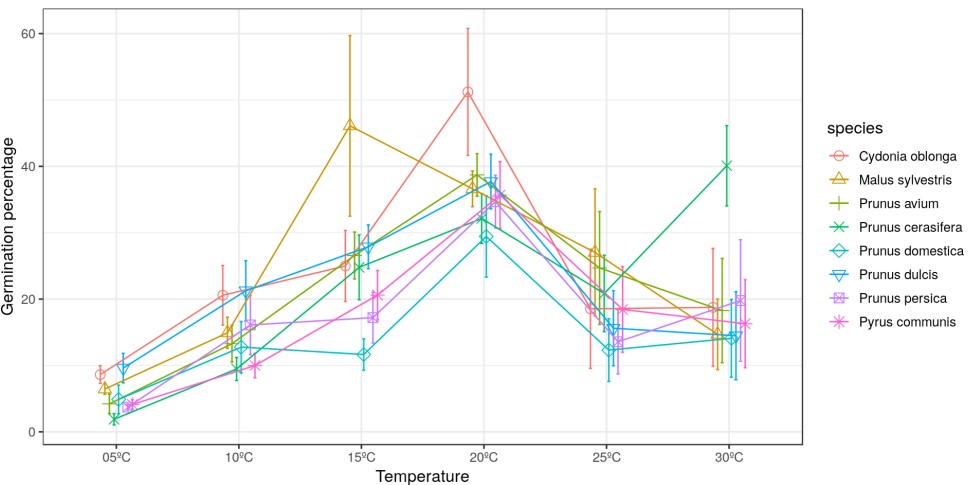

**Figure 1** **Effect of temperature on the percentage of germinated pollen grains.**

the pollen had germinated when the length of the pollen tube exceeded the diameter of its pollen grain. The parameters mentioned above were measured for the first 100 pollen grains observed in each plate. In case of finding a smaller number of grains, the parameters were calculated for the total grains observed.

All statistical analyses were executed using R (*R Core Team, 2017*) and RStudio (*RStudio Team, 2016*). ANOVA and Tukey post hoc tests were used to compare between treatments (temperature and freexing times) using "agricolae" package (*Mendiburu, 2019*). When significant differences were found, Levene's test and eta-squared statistics were calculated to assess the homogeneity of variances and the size effect in the ANOVA, respectively. Normality of residuals was tested using Shapiro–Wilk test and looking at density curves. Package "ggplot2" (*Wickham, 2016*) was used to plot the graphics.

## RESULTS

The maximum pollen germination ranged between 30–52% for all studied species. *Malus sylvestris* achieved its maximum germination (46%) at 15 °C. *Cydonia oblonga*, *Pr. avium*, *Pr. domestica*, *Pr. dulcis*, *Pr. persica* and *Py. communis* presented their maximum germination at 20 °C. On the other hand, *Pr. cerasifera* reached its maximum germination (40%) at 30 °C (Fig. 1).

Significant differences were found for the germination percentages between temperatures only in quince and apple (Tables 2 and 3). For *C. oblonga,* significant differences were observed between the percentage of grains that germinated at 20 °C and those that germinated at 5–10 °C and 25–30 °C (Table 2). For *M. sylvestris*, significant differences were observed between 15 °C and the temperatures of 5 °C, 10 °C and 30 °C (Table 3).

The longest pollen tube length was obtained in apple tree, with an average length of 2.5 times the diameter of pollen grain at 10 °C, while no significant differences were observed between temperatures for pollen tube length. Differences in *M. sylvestris* were seen only between the mean pollen tube length at 10 °C and 25 °C, while the highest values were

**Table 2** ANOVA analysis of the effect of temperature on the percentage of germinated pollen grains in *Cydonia oblonga*.

| Temperature | N | Mean | HSD | sd | se | skew | kurtosis | Shapiro |
|---|---|---|---|---|---|---|---|---|
| 5 °C | 8 | 8.627595 | b | 3.7653 | 1.33123 | −2.0485 | 4.9051 | 0.00934 |
| 10 °C | 9 | 20.575099 | b | 13.4543 | 4.48475 | 1.3232 | 2.2126 | 0.16584 |
| 15 °C | 9 | 24.999794 | ab | 16.0393 | 5.34642 | 2.7150 | 7.7177 | 0.00008 |
| 20 °C | 9 | 51.217313 | a | 28.7425 | 9.58084 | 1.2963 | 0.2574 | 0.01005 |
| 25 °C | 7 | 18.537415 | b | 23.7560 | 8.97891 | 0.5926 | −1.9428 | 0.01457 |
| 30 °C | 7 | 18.753307 | b | 23.4985 | 8.88160 | 0.4113 | −2.6631 | 0.00517 |
| | Df | Sum sq | Mean sq | F value | Pr(>F) | eta.sq | Levene | Shapiro |
| Temperature | 5 | 9060.72 | 1812.1440 | 4.60706 | 0.00187 | 0.34883 | 0.00103 | NA |
| Residuals | 43 | 16913.66 | 393.3409 | NA | NA | NA | NA | 0.00026 |

Notes.

   *N*, number of repetitions; HSD, honestly significant difference, different letters mean significant differences for alpha = 0.55; sd, standard deviation; se, standard error.

**Table 3** ANOVA analysis of the effect of temperature on the percentage of germinated pollen grains in *Malus sylvestris*.

| Temperature | N | Mean | HSD | sd | se | skew | Kurtosis | Shapiro |
|---|---|---|---|---|---|---|---|---|
| 5 °C | 9 | 6.447795 | b | 2.3557 | 0.7852 | 0.186818 | −1.2641 | 0.67991 |
| 10 °C | 9 | 14.920536 | b | 7.0519 | 2.3506 | 0.44042 | −0.7126 | 0.72167 |
| 15 °C | 9 | 46.103896 | a | 40.8139 | 13.6046 | 0.78479 | −1.7173 | 0.00207 |
| 20 °C | 9 | 36.612823 | ab | 8.0843 | 2.6947 | 0.64376 | 0.9319 | 0.46971 |
| 25 °C | 9 | 27.022669 | ab | 28.7423 | 9.5807 | 0.28891 | −2.0682 | 0.02976 |
| 30 °C | 9 | 14.682959 | b | 15.9491 | 5.3164 | 0.63759 | −1.7144 | 0.01341 |
| | Df | Sum sq | Mean sq | F value | Pr(>F) | eta.sq | Levene | Shapiro |
| Temperature | 5 | 10202.32 | 2040.465 | 4.27038 | 0.00271 | 0.307877 | 0 | NA |
| Residuals | 48 | 22935.26 | 477.8179 | NA | NA | NA | NA | 0.00245 |

Notes.

   *N*, number of repetitions; HSD, honestly significant difference, different letters mean significant differences for alpha = 0.55; sd, standard deviation; se, standard error.

obtained between 10 °C and 20 °C depending on the species. Apple tree was noteworthy, which reached the maximum pollen tube length at 10 °C. No significant differences were observed between temperatures for the maximum pollen tube length, except for *M. sylvestris* with differences between the maximum pollen tube length at 10 °C and at 25 ° C and for the mean tube length. The maximum pollen tube lengths for all the species were obtained at temperatures between 10 °C and 20 °C (Figs. 2 and 3).

Most of the species studied presented the lowest germination percentage (between 8% and 20% germination) in Treatment 2 (freezing time of four months). In contrast, *Pr. cerasifera* and *Py. communis* gave their lowest pollen germination percentages (21% and 12.5%) in Treatments 1 (freezing time of two months) and 3 (freezing time of six months), respectively. For *Pr. avium*, the lowest percentages were obtained in Treatments 2 and 3, both with about 17% germination. The lowest values were obtained by *Pr. persica* and *Pr. domestica* in Treatment 2, both with a germination percentage of 10% or lower. The highest value went to *C. oblonga* in Treatment 3, with more than 30% germination (Fig. 4).

All the species obtained their greatest average pollen tube length (1.0 to 2.4) and maximum pollen tube length (1.9 to 3.5) in Treatment 1 (Figs. 5 and 6, respectively). For

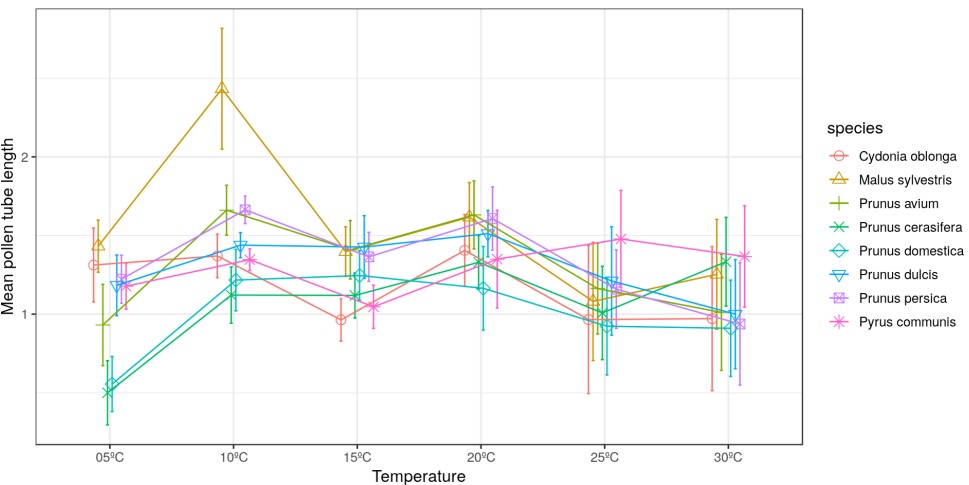

**Figure 2**   Effect of temperature on the average pollen tube length.

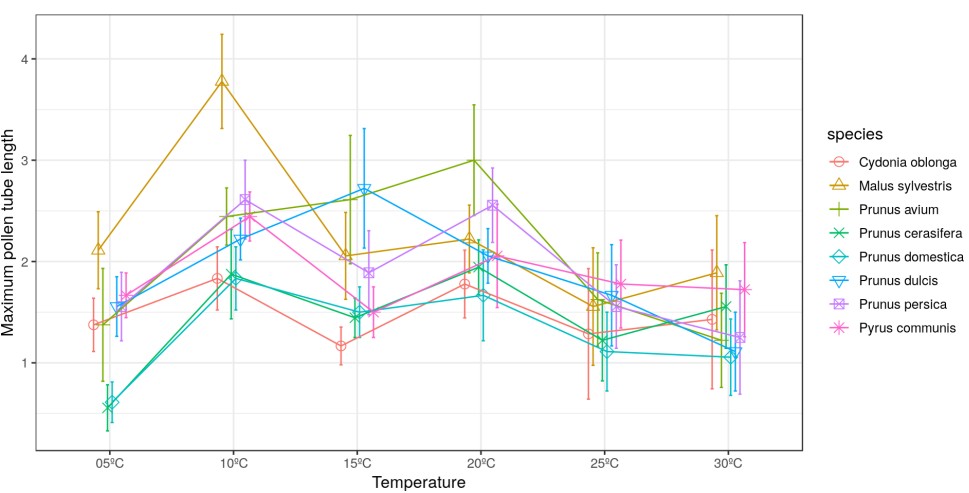

**Figure 3**   Effect of temperature on the maximum pollen tube length.

*M. sylvestris* and *Pr. avium*, the maximum pollen tube length recorded in Treatment 1 exceeded the pollen grain diameter by more than 3-fold. Likewise, all the species except *Pr. cerasifera* reached the shortest average and maximum lengths in Treatment 3. In this treatment (freezing time of six months), no maximum pollen tube length exceeded the pollen grain diameter by 1.5-fold.

Finally, the pollen germination percentage separated by species and treatments was represented (Fig. 7). A wide variability between both treatments and species was observed. Thus, it can be stated that the pollen germination percentage did not follow a clear pattern in the three studied treatments (freezing time).

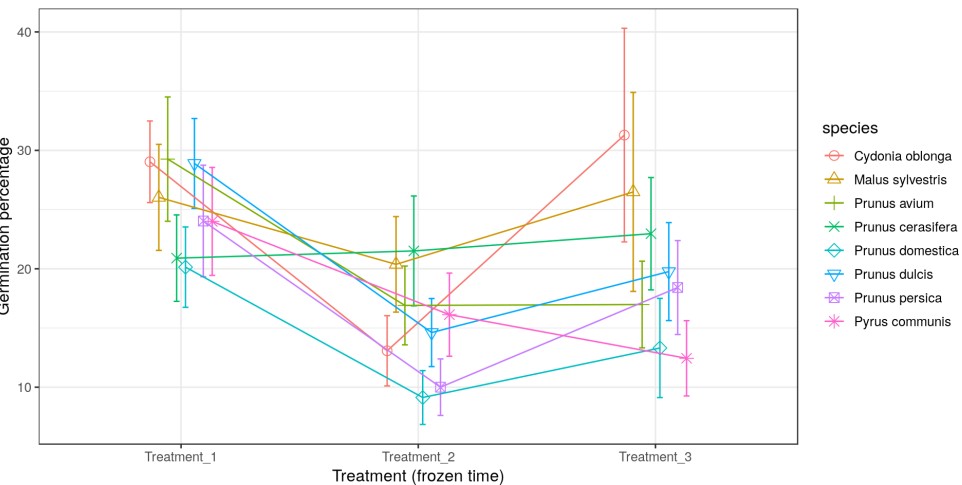

**Figure 4  Effect of freezing time on the pollen germination percentage.**

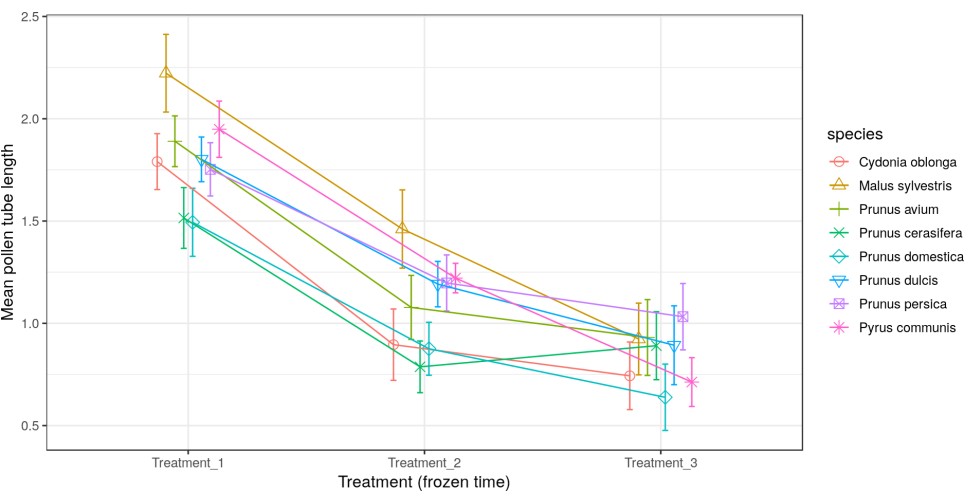

**Figure 5  Effect of freezing time on the average pollen tube length.**

## DISCUSSION

Low germination percentages were obtained for all the species at 5 °C and 10 °C. *Sanzol & Herrero (2001)* have already indicated how low temperatures slow down pollen growth in several species, such as pear or cherry. The present study verified that the optimum germination temperature for the studied *Rosaceae* species was variable. *M. sylvestris* was the species with the highest germination at the lowest temperature (15 °C), which can be accounted for by this tree originating from temperate areas (Agustí, 2010). The results obtained for *Pr. cerasifera* (higher germination at 30 °C) can also be explained by them originating from warmer climate areas (*Hegedűs & Halasz, 2006*). For the other *Prunus* species, the obtained results were intermediate and between previous cases, with an

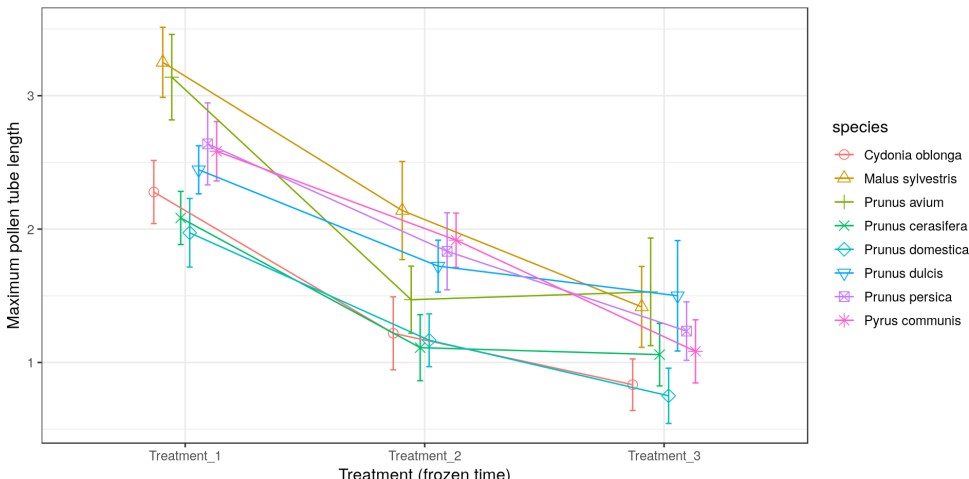

**Figure 6** **Effect of freezing time on the maximum pollen tube length.**

optimum germination temperature of 20 °C, which corresponds to this genus' origin from regions with an intermediate climate (Agustí, 2010). The results for *Malus*, *Prunus* and *Pyrus* coincided with those reported by other studies. *Hedhly, Hormaza & Herrero (2004)* observed how the pollen grain germination percentage in two cherry varieties lowered as temperature rose and obtained the highest germinated grain percentage at 20 °C. Other studies have observed that the optimum temperature for almond and peach pollen germination ranged between 16 °C and 23 °C (*Weinbaum, Parfitt & Polito, 1984*; *Hedhly, Hormaza & Herrero, 2005*). Also, *Sorkheh et al. (2018)* verified how the optimal temperature for pollen germination in various almond genotypes ranged between 20–25 °C. In contrast, *Sharafi (2011b)* found some variability in the temperature at which the highest germination percentage was recorded for several cultivars of almond, cherry, plum, apricot, prune (*Prunus salisina* L.) and sour cherry (*Prunus cerasus* L.).

In most evaluations, the obtained germination percentages were lower than 50%. *Bolat & Pirlak (1999)* tested different sucrose concentrations in their culture media on *Pr. armeniaca*, *Pr. avium* and *Pr. cerasus*, and obtained germination percentages below 50%. *Kakani et al. (2005)* measured 20–60% of pollen germination for *Gossypium* fresh pollen within a temperature range from 10 °C to 45 °C. *Hedhly, Hormaza & Herrero (2005)* obtained germination percentages of 80%–100% in peach fresh pollen for incubation temperatures like those herein tested. However, *Weinbaum, Parfitt & Polito (1984)* used pollen grains stored at −20 °C and did not specify the total storage time before they were used. These authors recorded 100% peach pollen germination when incubated at 20 °C and almost 99% almond pollen germination when incubated at 12 °C.

Wide variability has been observed in the results obtained in relation to the mean and maximum pollen tube lengths. *Weinbaum, Parfitt & Polito (1984)* observed that the pollen tube lengths of almond and peach did not vary despite testing a wide range of temperatures. *Sharafi (2011a)*, *Sharafi (2011b)* and *Sharafi (2011c)* reported differences for pollen tube lengths in peach, almond, cherry, plum, apple, pear and quince.

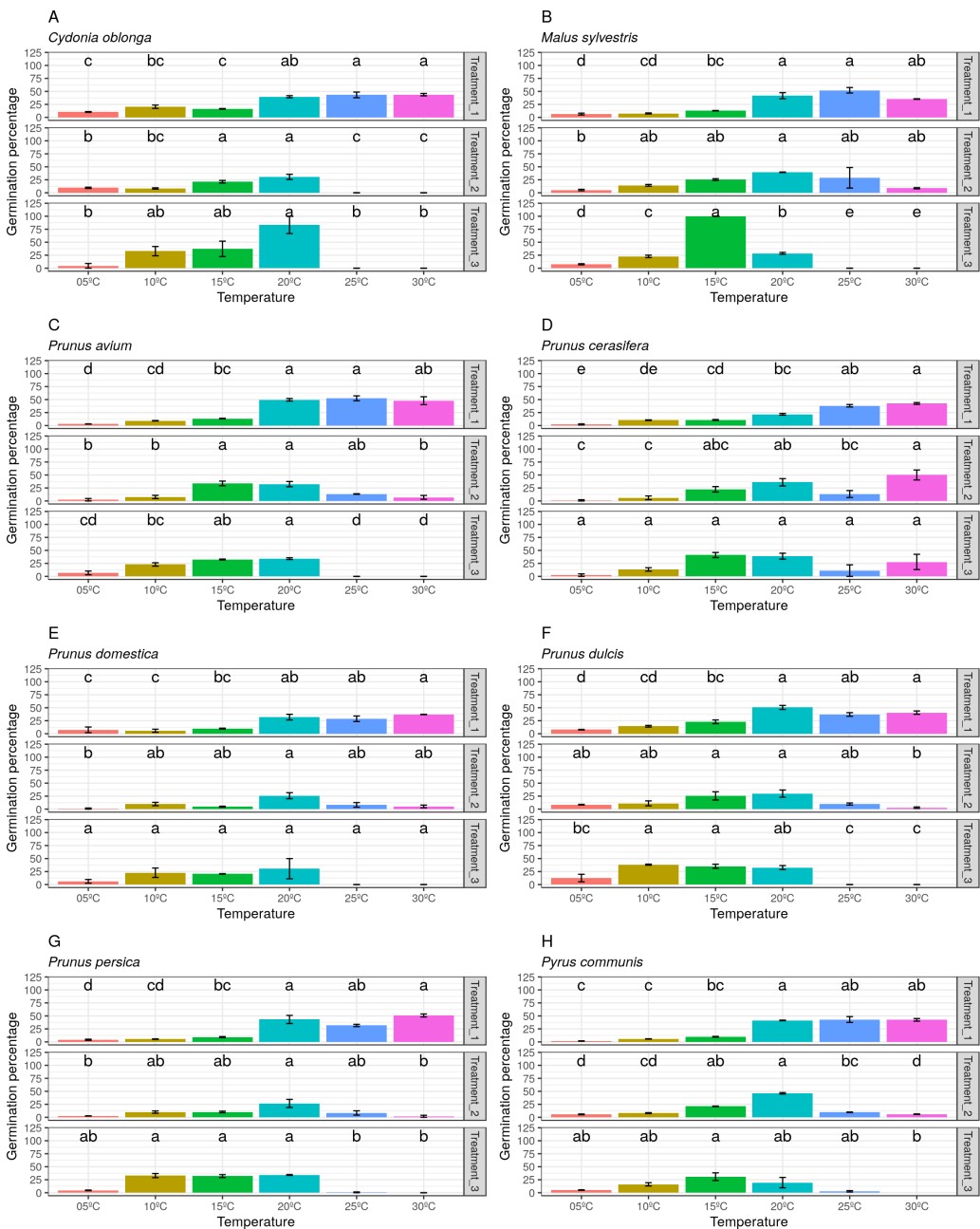

**Figure 7** **Germination percentage by temperatures and treatments (freezing times) for all species.**
(A) Data obtained in *Cydonia oblonga*. (B) Data obtained in *Malus sylvestris*. (C) Data obtained in *Prunus avium*. (D) Data obtained in *Prunus cerasifera*. (E) Data obtained in *Prunus domestica*. (F) Data obtained in *Prunus dulcis*. (G) Data obtained in *Prunus persica*. (H) Data obtained in *Pyrus communis*.

In this study, the mean pollen tube length of the five studied *Prunus* species oscillated within a wide range of temperatures (10 to 25 °C). In other *Prunus* species, like *Pr. cerasus*, the longest pollen tube length was observed between 15 °C and 20 °C (*Cerovic & Ruzic*,

*1992*). In other botanical families, similar results have been found. *Sedgley (1977)* observed that the pollen tube of avocado progressively grew between 12–17 °C and 29–33 °C.

*Hedhly, Hormaza & Herrero (2004)* used conserved pollen at −20 °C but did not specify the total storage time used under these conditions. *Weinbaum, Parfitt & Polito (1984)* retained pollen at −20 °C until further use. In all these cases, the obtained results do not seem to indicate that freezing at −20 °C modified normal pollen germination. Other studies carried out on species from other botanical families have indicated that the germinative capacity of freezing pollen is not necessarily worse than that of fresh pollen. For example, *Khan & Perveen (2006a)* indicated that *Ab. esculentus* pollen conserved at −20 °C germinated better than the pollen conserved at 4 °C and fresh pollen. The same authors observed that *S. melongena* pollen conserved at −20 °C germinated better than fresh pollen (*Khan & Perveen, 2006b*) and that *Prae. fistulosus* pollen maintenance improved at low temperatures (−20 and −30 °C) compared to fresh pollen (*Perveen & Khan, 2011*).

Those studies that have attempted to improve the pollen germination methodology have left aside storage conditions when using fresh pollen: different sucrose concentrations (Borlat and Pirlak 1999) and distinct media and temperatures for *Arabidopsis thaliana* (L.) Heynh. (*Boavida & McCormick, 2007*), cotton (*Kakani et al., 2005*) or *Pr. persica* (*Hedhly, Hormaza & Herrero, 2005*).

Several studies have indicated that freezing fresh flowers at low temperatures is a suitable way to preserve pollen grains in the long term. *Khan & Perveen (2014)* verified that the viability of pollen from several *Citrus* species remained at around 50–60% after four months at −20 °C. *Perveen & Khan (2008)* pointed out that *M. pumila* pollen maintained 65% germination capability for four months at −60 °C, but germination capability dropped to 16% for this same species at −20 °C.

*Towil (2010)* reported that pollen storage between −10 °C and −20 °C can be used to conserve material in the very long term; e.g., one to three years. However, this result should be qualified according to species. Thus, the pollens of *Citrus grandis* (L.) Osbeck and *Citrus medica* L. maintained their germination capacity for three years at −20 °C, while the germination of some grasses like *Lolium multiflorum* Lam. or *Pennisetum glaucum* (L.) R.Br., diminished by less than two months at the same temperature. However, the above-cited author indicated that the germination percentage of some Rosaceae, such as *Pr. domestica,* remained close to 60% after 2.5 years at −20 °C, while *P. persica* achieved germination percentages higher than 65% with storage times between four and nine years at −20 °C.

In this study, the germination parameters did not follow a logical pattern between Treatment 1 (two months stored at −20 °C), Treatment 2 (four months stored at −20 °C) and Treatment 3 (six months stored at −20 °C). In the case of the pollen tube length, average and maximum length were decreasing from Treatment 1 to Treatment 3. The storage at low temperatures could modify the pollen response to stress situations. *Towil (2010)* commented that the pollen storage between −10 and −20 °C could lose quality for the long-term storage. So, this observed variation may be due to a decrease in pollen quality as storage time passes. But the pollen germination percentage obtained in the three treatments did not follow a logical pattern. Between Treatment 1 and Treatment 2 there is

a decrease in the pollen germination percentage, which coincides with the observed pollen tube length data. Nevertheless, we have observed an increase of the pollen germination percentage for all the studied species, except *Py. communis*, between Treatment 2 and Treatment 3. This change in trend does not correspond to the observed results of pollen tube length between Treatment 2 and Treatment 3. Consequently, the use of pollen from *C. oblonga*, *M. sylvestris*, *Pr. avium*, *Pr. cerasifera*, *Pr. domestica*, *Pr. dulcis*, *Pr. persica* and *Py. communis* stored at −20 °C for a period of six months could influence the results.

We have observed that the pollen used in much of the studies carried out about the effect of temperature on the germination of pollen has been stored in freezing. It is amply proven that low conservation temperatures do not affect the germination capability of pollen. For this reason, the storage at low temperatures is the most widespread conservation method. Anyway, the different germination patterns observed in our study make us have doubts about their reliability.

So, the freezing temperature and the total freezing time are two factors that should not be taken lightly. Freezing Rosaceae pollen at −20 °C is a suitable method for long-term preservation, but it may not be for a reliable study on the effect of temperature. It is easy to assume that the results made on fresh pollen subjected to a temperature scale are directly extrapolated to natural conditions, but it is no longer clear whether those same results obtained when using frozen pollen could be extrapolated with the same reliability. In short, the data we have collected may be important for further studies on pollen germination. Perhaps we should rethink what is the most effective way to store pollen depending on what we are going to use that pollen later. In any case, it would be necessary to compare the germination percentage of pollen preserved at −20 °C against fresh pollen in these Rosaceae species.

## CONCLUSIONS

The results obtained herein confirmed that freezing Rosaceae pollen at −20 °C is a suitable method for its long-term preservation. The pollen of all the studied species maintained its germinative capability throughout the six months in which it was stored at −20 °C. However, the pollen germination pattern changed depending on the months in which the pollen remained frozen. The results of the three parameters measured (pollen germination, average pollen tube length and maximum pollen tube length) were different in the three treatments studied (two, four and six months stored at −20 °C). One explanation could be that storage at low temperatures could modify the pollen response to stress situations. Consequently, studies that aim to assess the effect of germination temperatures should use fresh pollen. On the other hand, it has been observed that the average pollen tube length and maximum pollen tube length are not suitable evaluation parameters due to their wide variability.

### Funding

This work was supported by the Asociación Club de Variedades Vegetales Protegidas as a part of a project with the Universitat Politècnica de València (UPV 20170673). The funders had no role in study design, data collection and analysis, decision to publish, or preparation of the manuscript.

### Grant Disclosures

The following grant information was disclosed by the authors:
Asociación Club de Variedades Vegetales Protegidas as a part of a project with the Universitat Politècnica de València (UPV 20170673).

### Competing Interests

The authors declare there are no competing interests.

### Author Contributions

- Roberto Beltrán conceived and designed the experiments, performed the experiments, analyzed the data, prepared figures and/or tables, authored or reviewed drafts of the paper, approved the final draft.
- Aina Valls performed the experiments, prepared figures and/or tables, approved the final draft.
- Nuria Cebrián performed the experiments, contributed reagents/materials/analysis tools, prepared figures and/or tables, approved the final draft.
- Carlos Zornoza performed the experiments, contributed reagents/materials/analysis tools, authored or reviewed drafts of the paper, approved the final draft.
- Francisco Garcia Breijo and José Reig Armiñana conceived and designed the experiments, contributed reagents/materials/analysis tools, authored or reviewed drafts of the paper, approved the final draft.
- Alfonso Garmendia conceived and designed the experiments, analyzed the data, prepared figures and/or tables, authored or reviewed drafts of the paper, approved the final draft.
- Hugo Merle conceived and designed the experiments, analyzed the data, authored or reviewed drafts of the paper, approved the final draft.

### Data Availability

   The raw measurements are available as a Supplemental File.

### Supplemental Information

Supplemental information for this article can be found online at http://dx.doi.org/10.7717/peerj.8195#supplemental-information.

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
