# Peer review of "Effect of temperature on pollen germination for several Rosaceae species: influence of freezing conservation time on germination patterns"

_PeerJ, doi:10.7717/peerj.8195_

## Round 0.1 · original submission · Major Revisions

The manuscript presents an interesting account of testing the effects of temperature and cold storage on pollen germination. Considering the economic importance of the focal species these data may be useful in increasing fruit set or allowing farmers to better predict yield in a given season. The importance of the results was not stated as such in the manuscript, however. Thus, the reader is currently left wondering why this study is important and what it contributes to the field. As such, based on the comments from the reviewers and my own reading, I recommend several main areas for revision.

1) Introduction – expand the introduction somewhat. Why is it important to study pollen germination for this group? How might the results/information help? On lines 49-58 it is stated that understanding pollinators is important for increased fruit set, then it states that understanding pollen germination is important. How are these two points connected? Will understanding pollinators help in understanding pollen germination somehow? Even if pollinators increase pollination, how does this impact pollen germination? These are actually two separate issues (as I am sure you are aware) that need to be more clearly explained. Why are freezing temperatures of particular interest? These types of important connections aren’t yet being made for the reader.

2) Methods - the reviewers expressed concern regarding the pollen grain germination. Was a specific threshold considered in order to declare that a pollen grain had germinated? Were any controls used?

3) Discussion – a solid paragraph on why these data are important and how they contribute to agriculture/science would be a significant improvement.

4) Figures - standard error bars should be provided for the figures.

Additional comments from myself and the reviewers follow.

Lines 26-28 – because you are dealing with multiple genera beginning with P, you will need to provide the full genus name on the subsequent lines or provide an expanded abbreviation – such as Pr vs Py. Not everyone will be automatically familiar with the species within Prunus and Pyrus.

Lines 31-32 – within the abstract it will be easier to refer to the conditions rather than the block. For example, “…percentages were obtained under four months of freezing”

Line 47 – which P genus? Amend throughout the manuscript

Line 94 – show is a better word than reveal in this situation

Line 101 – would storage be a better term than conservation? Conservation tends to automatically make people think of saving something threatened, whereas this work was essentially storing pollen for a later date.

Line 105 and throughout – it is scientific convention to write numbers under 10 in full (i.e., six not 6)

Line 103-106 (and throughout) – it would be good to either refer to the treatments by times (i.e., 2, 4, 6 months) or by name with an explanation that you will hereafter refer to the treatments as Block-1 etc.

Line 109-111 – Unnecessary detail. Easier to just say ‘pollen grains were extracted and placed in 5 ml of BK media (DETAILS) to induce germination’. Provide the detail about the media from lines 113-115 in the parentheses.

Line 131 (and throughout) – at the beginning of sentences the full genus name must be used. Also, just at a lower temperature or the lowest temperature you tested?

Line 142 (and throughout) – when reporting an average like this it is helpful to also report the SE in parentheses.

Line 152-158 – what were the highest and lowest? Provide values within the text in addition to referring readers to the figures.

Line 190 (and throughout) – no space between a number and the percent sign

Discussion – multiple species are referred to that are not within Prunus etc. For every new species that also comes from a previously unmentioned genus you need to write the full scientific name. If these genera also have names starting with the same letter (i.e., A. something vs A. somethingelse) you need to have a modified abbreviation for the genus so that they can be distinguished as different species and genera. This has been done for some of the species referred to but not all. Please be consistent.

Line 245 – is it really ‘good’ or is it suitable? The data presented suggest it is suitable – as in, it is ok but if something better were found that would be good.


Reviewer 1 ·

Basic reporting

The manuscript that entitled: Effect of temperature on pollen germination for several Rosaceae species. Influence of freezing conservation time on germination patterns (#38121) carefully read and could help researcher and fruit breeder to improvement of genotypes against environmental conditions (Abiotic stress) for example temperature which the author explored of the research on eight Rosaceae species.
Overall the article is suitable but the English language should be improved to ensure that an international audience can clearly understand your text. Some parts of the text in the manuscript unclear and need to be again checked it by one Speck -English native English around the whole of document.
The authors need to be introduce litterateur review on current references on Prunus species, which is not issued and need to be developed by recent reference of these studied species. For example:
Sorkheh et al. (2011) Influence of temperature on the in vitro pollen germination and pollen tube growth of various native Iranian almonds (Prunus L. spp.) species. Trees (2011) 25:809–822.
Sorkheh et al. (2011) Response of in vitro pollen germination and pollen tube growth of almond (Prunus dulcis Mill.) to temperature, polyamines and polyamine synthesis inhibitor. Biochemical Systematics and Ecology 39 (2011) 749–757.
Sorkheh et al. (2018) Interactive effects of temperature and genotype on almond (Prunus dulcis L.)
pollen germination and tube length. Scientia Horticulturae 227 (2018) 162–168.
The structure of Tables and Figures need to be minor revision for clearly understand your text. For example: Table 1 provides information on Latitude N, Longitude E, Elevation (m), Annual rainfall (mm) from each location that’s you are selected the species.
In Table 2, 3 and 4; please characterized each abbreviation as superscript in the Tables and so described them below of the Tables. This is importance clearly understand your text.
Please revised the title of Table 2: the percentage; Cydonia blonga

Figure 1-6 need to be added standard error bar for each point (temperature, block times). This is could help reader to clear find significant times for each species.

Experimental design

The research relevant suitable results and could helpful in fruit breeding. In this part suggested that’s calculated relative injury, pollen viability and so completed M&M on measurements. Methods need to be completed on details.

Validity of the findings

The validity is good and applied using statistics software. But the authors when working of pollen germination and pollen tube under temperature need to be clarified. For example; Relative injury (RI) to cell membranes resulting from the temperature treatments was calculated according to Kakani et al. (2002) and Acar and Kakani (2010).
Pollen viability; this index is very significant and need to be authors of the paper described of it under different treatments.
I would be like know the authors classified the species according to Sorkheh et all (2011) and Kakani et al. (2005) as tolerant, moderate and susceptible species. Because it’s very importance for fruit breeding and development of tolerant genotypes for different environmental conditions.

Additional comments

The manuscript that entitled: Effect of temperature on pollen germination for several Rosaceae species. Influence of freezing conservation time on germination patterns (#38121) carefully read and could help researcher and fruit breeder to improvement of genotypes against environmental conditions (Abiotic stress) for example temperature which the author explored of the research on eight Rosaceae species.
Overall the article is suitable but the English language should be improved to ensure that an international audience can clearly understand your text. Some parts of the text in the manuscript unclear and need to be again checked it by one Speck -English native English around the whole of document.
The authors need to be introduce litterateur review on current references on Prunus species, which is not issued and need to be developed by recent reference of these studied species. For example:
Sorkheh et al. (2011) Influence of temperature on the in vitro pollen germination and pollen tube growth of various native Iranian almonds (Prunus L. spp.) species. Trees (2011) 25:809–822.
Sorkheh et al. (2011) Response of in vitro pollen germination and pollen tube growth of almond (Prunus dulcis Mill.) to temperature, polyamines and polyamine synthesis inhibitor. Biochemical Systematics and Ecology 39 (2011) 749–757.
Sorkheh et al. (2018) Interactive effects of temperature and genotype on almond (Prunus dulcis L.)
pollen germination and tube length. Scientia Horticulturae 227 (2018) 162–168.
The structure of Tables and Figures need to be minor revision for clearly understand your text. For example: Table 1 provides information on Latitude N, Longitude E, Elevation (m), Annual rainfall (mm) from each location that’s you are selected the species.
In Table 2, 3 and 4; please characterized each abbreviation as superscript in the Tables and so described them below of the Tables. This is importance clearly understand your text.
Please revised the title of Table 2: the percentage; Cydonia blonga

Figure 1-6 need to be added standard error bar for each point (temperature, block times). This is could help reader to clear find significant times for each species.

The research relevant suitable results and could helpful in fruit breeding. In this part suggested that’s calculated relative injury, pollen viability and so completed M&M on measurements. Methods need to be completed on details.


The validity is good and applied using statistics software. But the authors when working of pollen germination and pollen tube under temperature need to be clarified. For example; Relative injury (RI) to cell membranes resulting from the temperature treatments was calculated according to Kakani et al. (2002) and Acar and Kakani (2010).
Pollen viability; this index is very significant and need to be authors of the paper described of it under different treatments.
I would be like know the authors classified the species according to Sorkheh et all (2011) and Kakani et al. (2005) as tolerant, moderate and susceptible species. Because it’s very importance for fruit breeding and development of tolerant genotypes for different environmental conditions.

The manuscript have suitable results and could be accepted /published in Peer J. after revision.

Annotated reviews are not available for download in order to protect the identity of reviewers who chose to remain anonymous.

·

Basic reporting

It is interesting to read the article and I would like to thank the authors for their work. The authors have evaluated how temperature and freezing time affect pollen germination and mean and maximum pollen tube length in 9 species of Rosaceae. They show that different species have different optimums and that in general terms freezing time was negatively correlated with pollen tube length (showed graphically). However, there are some major concerns after reading the manuscript that need to be addressed.

The English needs more work, I’m not native English speaker so I cannot comment much on it, but some phrases are difficult to understand. There are some references from Spanish journals, this makes difficult to international readers to follow them if necessary, I suggest to find these references in English. There are also phrases that need referencing. Moreover, there are some citations that need better explanation in the text. The text need to be clearer and everything better explained. I cannot comment on the literature much because I’m not very familiar with it. However, I have added few suggestions below.

The tables and figures need more work. Improve axis labels, italicise species names, improve captions… In example, Figure 7 has bars that are not described, what it is, confidence intervals, sd, sem? There are just 3 tables with stats results and there are 9 species. It seems that is based on the significance of the results. I suggest here to make a table with all the species just showing that there are differences or not with the Anova output and maybe add Tukey test but no need to report all the basic statistics, they are informative before the analysis to understand the data but not as main result of the study. If you do believe that is necessary to show these data, I think that showing it graphically will make it easier for the reader.

Moreover, the analysis just tests differences of pollen germination rate with different temperatures. Why not on mean and maximum pollen tube length? Compare it statistically too. How do you know if freezing time and temperature is affecting the different proxies, if you don’t have a control? How can we know that the selected temperature to store the samples is good or bad? Why not testing different freezing temperatures too? The comparisons of the effect of temperature are with the same freezing times or different or grouping all of them? This is very relevant and should be clearly specified.

Here I add some comments in detail about the abstract and the introduction. Methods, results and discussion are commented on sections 2 and 3.

Abstract

The abstract lacks an intro to your research. State your questions and highlight your findings. What do you test? Lack of consistency with the scientific names. Instead of name the freezing times “blocks”, treatments sound better for me, at the end you are testing that, different treatments. The underscore of block, for the r script makes sense but not for a paper presentation. So, what about state your results like: Pollen germination was maximum for all species between 15 and 30 degrees. For xxx species maximum pollen germination was… Germination did not change significantly for xxx species with freezing time but we found significant …

Line 22: Even though you mention that you have used eight species here, but in the methodology section you tested nine species including E. Japonica? Therefore, please check this and change it accordingly.

Introduction

The first paragraph of the introduction should be catchier… You already describe what your species are in table 1. If you want to mention the species, maybe in the last paragraph of the intro or in methods you can describe them with more detail. You need to introduce the topic to your reader in a more elegant way.

For me is lacking that you state clearly what the knowledge gap is and how this research can help in the field of pollen biology. For example, knowing the correct store methodology can help breeders or research of pollen biology to be performed at any time during the year without reduction in success of pollen germination… I’m lacking this on the text, the practicality of it and more exhaustive description of what we do know and do not know.

The second paragraph of the intro I think is find there just need to be improved and replace the Spanish reference for an English one. Ref for Rosaceae flowers are actinomorphic and hermaphrodite. Sargent 2003 will do the work for the symmetry. You should find other ref for compatibility. Scientific names! Make it consistent.

Third paragraph, if you talk about the studies, mention what they have done and found. So, we will know what past literature has found and what needs to be addressed. Could be interesting to link the different temperature ranges of the species with the maximum pollen germination and pollen tube length. I would consider to move the first paragraph of methods to the intro and develop this idea as a research question that could help the paper to grow.

Fifth paragraph, different temperatures are tested in the different studies, what did they find? Any temperature is better? If so, explain it. Rajasekharan & Ganeshan 1994, journal of Horticultural Science showed that with liquid nitrogen, cryostored pollen maintained its ability to produced seeds on roses. This could help to build this paragraph… Why you test just one temperature? Is any reason? Nath & Anderson 1975 Cryobiology, talk about the cooling rate… As a variable that affect pollen germination, I’m not very familiar as I said with the literature but everything should be expanded and more variables consider.

Last paragraph of the intro, you don’t test temperature conservation conditions… Build your research question with the correct terminology.

Experimental design

I think that the article fits the aims and scope of the journal and I I do believe, that their research can help the field of pollen biology. However, their research questions are not clearly tested and their methodology lacks detail to make this study reproducible.

The methods section needs more detail:

I will not use E. japonica due to is from 2017 and not 2018 like the rest… Better to avoid the lack of consistency to have less noisy results.

How did you storage the samples? Tube, bag?

Again, don’t call it block call it treatment.

Delete lines 99 to 102. Don’t include loquat. This makes your research inconsistent.

The analysis needs better explanation and structure. If you compare the differences with Anova’s between pairs with Tukey test, explain it together, don’t say that you use Tukey at the end with the CRAN packages. It’s good that you do basic descriptive stats for understanding your data but I won’t mention them in methods. You can show that graphically with means and confidence intervals and is more elegant that a table full of numbers in case you really want to add them.

Please mention sample size in the text. Why in the table of results is a different N for each temperature. Any reason of the different sample sizes?

How do you consider that a pollen grain has germinated or not? Any minimal length? We should help future work to make it reproducible. And please we need the units and how you measured it. From all the pollen grains which ones you counted or measured? all? subset?

The results are very chaotic, is important to make the life easier to the reader. Divide your results in your research questions and reference figures and analysis outputs.

Seems that pollen tube maximum and average length decrease with freezing time as seen in the figures. This is interesting! Sell it. However, the percentage of germinated pollen grains seems to don’t change that much. We need stats to confirm this, add them. Again, instead of calling it block, “Treatment” sounds better and remove underscore.

Validity of the findings

The hypothesis need to be clearly tested, at the moment the discussion is lacking substance. Moreover, the conclusions don't fit the results. How it can be the conclusion that conserving pollen at -20 degrees Celsius is good? You are not testing that, you are testing different freezing periods and different ranges of temperatures for germination and pollen tube length... I guess that is said because the germination rate is not low... But you should mention what you are really testing. The discussion needs more though, better explanation and going from the bigger picture to the specific detail.

Additional comments

Other comments:

Why pollen tube length decreased with freezing time but no germination? I'm lacking more discussion about this.

I like the idea that the results are trying to be matched with the climatic regions this could be converted to a research question and tested explicitly.

Figure 1 Y axis, maybe better “Germination rate”? I guess is an average, so describe it in the caption.

In the Figures 1 to 6 explain why you have a line that links the points… Visualization purposes?

Line 40: after worldwide “like”

Line 51: Why they behave differently if they are self-compatible or self-incompatible. Explain it.

Line 59 to 67: So, you say there are several studies with Rosaceae but what do they say or conclude? Is that relevant to your research?

Line 68/69: Linked with the last paragraph (59/67), now you talk about more studies are the same ones? Reference them.

Line 86 to 88: So, you test pollen germination with different temperatures and different freezing times? What about pollen tube length?

Line 91: Rosaceae family species, fix English.

Line 92: Fix English and add reference to the temperature range.

Line 110/112: Rephrase.

Line 118/119: Fix English.

Line 124: E#ect, fix.

---

## Round 0.2 · Minor Revisions

The clarity and content of the manuscript has been much improved. However, the introduction is still missing the big picture statement. Why should anyone care what the best pollen germination conditions are for these species? Are there inherent problems with fruit/seed set for commercial markets? Are there pollen-pistil incompatibilities among varieties or individuals that might limit fruit/seed set? Will climate change alter the efficacy of fertilization based on temperature effects on pollen tube length? Why might we need to store pollen for these species and what does it mean if pollen performs more poorly after freezing? The reader really needs this type of context to become truly interested in the paper. Perhaps this information is inherent to someone deeply immersed in pollen ecology but for many readers it will not be.

In addition, there are several minor typos:
Line 58 – change genus to the plural form genera
Line 143 – ANOVA
Line 146 – it looks like something happened to the text formatting for the word ‘effect’, the ‘ff’ appears as a box
Line 152 – use the full genus name at the beginning of a sentence

Reviewer 1 ·

Basic reporting

The manuscript is revised based on the comments of respectful reviewers and I think the manuscript could be considering for publication in PeerJ.

Experimental design

The experimental information on M&M are suitable and with sufficient detail and information.

Validity of the findings

the validity of findings have done using statistics but so important parameters that's completed the work and could be beneficent for fruit breeding Relative injury (RI) and pollen viability, which the authors have not calculated.

Additional comments

The manuscript has suitable information on effect of temperature on pollen germination for several Rosaceae species this is very important for next experimental that's could help fruit breeder in the selection of tolerant and sensitive cultivars for different environmental conditions. Thus, the manuscript could be considering for publication in PreeJ.

---

## Round 0.3 · Minor Revisions

The introduction has been much improved. However, there are some typos and grammatical errors introduced with the improvements.

Line 64 - new paragraph
Line 76 - avocado is its own cultivation of warm areas that is already being cultivated - this doesn't make sense at the moment. Perhaps cultivated is repeated in too many places?
Line 79 - excess heat during flowering and fertilization conditions the harvest yield - missing words??

---

## Round 0.4 · accepted · Accept

We look forward to publishing your work!